# FULE—Functionality, Usability, Look-and-Feel and Evaluation Novel User-Centered Product Design Methodology—Illustrated in the Case of an Autonomous Medical Device

**Ela Liberman-Pincu** [1] and **Yuval Bitan** [2,3,*]

1   Department of Industrial Engineering and Management, Ben-Gurion University of the Negev,
    Be'er-Sheva 8410501, Israel; elapin@post.bgu.ac.il
2   Department of Health Systems Management, Ben-Gurion University of the Negev, Be'er-Sheva 8410501, Israel
3   Department of Mechanical & Industrial Engineering, University of Toronto, Toronto, ON M5S 1A1, Canada
*   Correspondence: yuval@bitan.net

**Abstract:** The overall goal of the novel Functionality, Usability, Look-and-Feel, and Evaluation (FULE) user-centered methodology for product design proposed in this paper is to develop usable and aesthetic products. Comprising several product design methods, this novel methodology we devised focuses on the product designer's role and responsibility. Following the first three formative assessment phases that define the product's functionality, usability, and look-and-feel, the summative evaluation phase not only assesses the product, but also provide guidelines to its implementation, marketing, and support. A case study devoted to the design of an autonomous medical device illustrates how the FULE methodology can provide the designer with tools to better select among design alternatives and contribute to reducing bias and subjective decisions.

**Keywords:** design methods; design process; detailed design; new product development; FULE methodology; user-centered design; usability; aesthetics

## 1. Introduction

Whether involving a brand-new product or improving and updating an existing one to evolving and changing demands, the product design process is need-driven. New product innovation can be manifested as market-pull (filling customer needs including improving ergonomics and usability), technology-push (improving technological features such as accuracy, speed, etc. due to scientific breakthrough), or design-driven innovation (creating a new look-and-feel as part of a new branding process) [1].

Every product is unique; consequently, each design process requires different tools, advisors, and research methods relevant to its specific needs. A structured methodology hopefully leads to a structured process and an optimal product.

Numerous product management and development methodologies have evolved over the years to initiate an effective product design process that meets changing market needs and demands. Whereas engineering research and methodologies have existed since the 19th century, design became a topic of research only towards the end of the 20th century [2]. Some of the methodologies focus on a narrow part of the process, aiming to generate innovative solutions and produce a great number of ideas and concepts [3,4]. These include methods for exchanging knowledge in multidisciplinary product development teams [5], solving ergonomic design and usability issues [6], examining the aesthetic aspects of the product [7], or focusing on the design of particular objects and demands such as soft wearable devices, sports equipment, or emergency ventilators [8–10]. Other methods take a more holistic approach and observe the process throughout its length and breadth [11]. We developed and formulated the Functionality, Usability, Look-and-Feel, and Evaluation (FULE) methodology based on ten years of experience in developing various medical devices and participating in many design processes. This method evolved

from the necessity to create a structured methodology that focuses on the designer's role and responsibility in the medical system product development process. More than other design fields, medical device design presents significant challenges related to safety, ease of use, and appearance. Nevertheless, industrial design is often perceived as redundant in the medical device development process that usually focuses on technological aspects.

While developers and engineers aim to optimize for functionality, the designer aims to optimize for user experience. These two goals must go hand in hand from the beginning of the process to achieve success. FULE methodology guides the designer in this complex multidisciplinary process and provides tools for better team communication. The need for this new methodology is derived from various obstacles encountered during the development process (including inadequate designer–team member communication, decision-making difficulties, unsuitable chosen aesthetics, etc.) that resulted in a waste of time and, hence, money. Using failure analysis at the end of every design project helped us analyze obstacles, find solutions, and determine rules. These were then applied in the next projects. The result was a clear guide that leads the designer through the whole procedure, starting at the beginning of a process and ending only after the product is already in use by real users. FULE structures the industrial designer's path to explore and develop design solutions step-by-step using continuous formative assessments in every phase with a final summative assessment that evaluates the process, the outcome, and the methodology for future adjustments. FULE provides visual means to discuss alternatives and conflict demands and helps the designer communicate with other team members.

This paper presents an overview of the FULE methodology phases, followed by a case study that illustrates implementation of the FULE methodology.

## 2. Background

In order to identify and understand the components needed for a successful user-centered product design process of a new medical device, first, we need to define medical devices and their users and understand the nature of their environment. We must also become familiar with other design methodologies and approaches and understand what the properties are of a design/development approach that should lead to the design of trustworthy products.

### 2.1. Medical Device Design

The FULE methodology is focused on the product design (the physical form of an object) of professional medical devices intended for utilization by professional medical teams (i.e., doctors, nurses) in professional medical environments (i.e., hospitals, clinics). In other words, complex environments, frequent staff turnover, and various users hold a wide range of experience.

The hospital work environment varies with each unit's unique characteristics, whether in the operating rooms, emergency rooms, intensive care units, etc. The nature of a new device's work environment influences the design at various levels, such as the degree of sterilization and disinfection required, the nature of alarms and indicators, and the design of interfaces. The intensive care unit (ICU), for example, is a highly challenging environment in which nurses deal with more than 500 tasks a day, in addition to administrative obligations and an information overflow both from IT systems (with a high rate of false alarms and irrelevant messages) and colleagues [12]. Nurses' high workload in ICUs leads to major worker stress and harms patient safety [13].

### 2.2. What is Good Design?

The shared goal of varying product development methodologies is to facilitate the development process in order to design successful products. The issue at hand is the definition of success in this respect, as is its evaluation.

The term success concerning product development could be related to various accomplishments, including product quality, cost and time to market, execution of technical needs,

the fulfillment of the perceived aesthetic, or ergonomic or societal needs [2]. Initiating a design process requires setting up respective goals and expectations. A research goal may refer to several factors, for example, (1) making a long and complex assembly process short and intuitive, or (2) using cheap disposable components—up to USD 2 per unit. All goals must be well defined and detailed, and they must have a hierarchy. A success criterion must be formulated for each factor. The criterion can be relative or absolute, qualitative, or quantitative. Defining a set of criteria during the preliminary stages is essential for the design process and influences the research approach and methods [2]. In terms of user-centered design, success would be the acceptance and adoption of the new system by the users, as well as good usability, ease of use, and clarity. In addition to these parameters, it is highly essential to gain users' trust, especially when dealing with autonomous systems.

### 2.3. Design Importance

Design plays an important role in creating trust in terms of both visibility and usability. The visual aesthetics of a product form the user's initial judgment (first impression). The elements affecting our first impression have been researched in many disciplines. It was found that consistent first impressions may be formed in the first 39 milliseconds based on visual information. In most cases, the first impression affects behavior, attitude, and relationship with a product in the mid- and long-term [14]. Studies suggest that even color alone has a psychological effect on the user, both on his emotions and his mood. This impact will differ from person to person, depending on various factors such as age, culture, etc. Color components seem to trigger the human arousal system and affect our perception and trust [15].

Having worked with a system, the user experiences its level of usability. Research indicates that trust in technology is supported or prevented by perceived usability. A low level of usability could result in a lack of trust and a negative attitude towards a product [16].

### 2.4. Summary

A literature overview suggests that designing a trustworthy medical device requires attention to specific features: functionality, usability, and visibility (look-and-feel). The FULE methodology we introduce in this study refers to the designer's roles and responsibilities when engaging in a new medical device's development process. Our methodology aims to design safe user-centered products in accordance with the user's physical and cognitive abilities.

### 3. Aim

The primary objective of this research is to define a new medical equipment design methodology: (FULE—Functionality, Usability, Look-and-Feel, Evaluation). FULE methodology was developed in order to help designers to design products with high usability and affordance. By applying this methodology, the design team is guided to reason every decision made in the design process. Thus, decisions are not biased by the subjective opinion of the designer. In addition, the methodology provides a basis to resolve conflicting demands and provides tools for discussion and alternative selection that leads to better product design.

In this paper, we illustrate some of the FULE phases and tools in several design projects. In addition, we present a case study of the full design process of a complementary autonomic breathing system by Hospitech® (Kfar Saba, Israel).

### 4. Study Structure

Aiming to develop a novel procedure that allows optimal design of a medical device, we first set out to define guidelines and principles for the FULE user-centered methodology. We then illustrate the FULE's advantages through a medical device case study where FULE was used to develop the second generation of a medical product that was complex and had

usability difficulties. As it is not feasible to evaluate a product development methodology through a controlled study, the case study serves both to illustrate an implementation of the methodology and to evaluate the FULE methodology through feedback that was collected from medical staff in two hospitals.

### 5. FULE Methodology

The FULE methodology applies several tools and methods designed for product designers. Unlike other methodologies, which look at the broad process and deal with varied aspects related to the development of a new product (e.g., business objective, budget, organizational, etc.), FULE focuses on the role and responsibility of the product designer during the development process of medical equipment. According to this methodology, the development process consists of four user-centered design phases of evaluation and design. The first three are formative assessment phases to define the future product in all three aspects (product functionality, usability, and look-and-feel). The final phase is a summative evaluation phase that may affect the product itself or provide product implementation, marketing, and support guidelines. Figure 1 provides an illustrative representation of FULE.

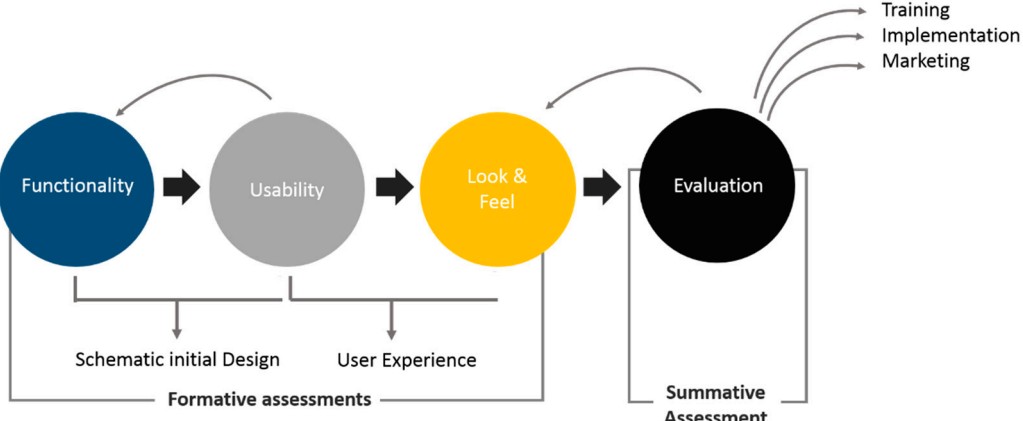

**Figure 1.** Illustrative representation of the Functionality, Usability, Look-and-Feel, and Evaluation (FULE) methodology. The development process consists of four user-centered design phases.

The FULE methodology is built like a pyramid with a clear hierarchy (as described in Figure 2). Each level is dependent on the layer below and sets a solid foundation for the next one, creating clear and concise requirement specifications of all aspects of the product: Functionality, Usability, and Look-and-Feel.

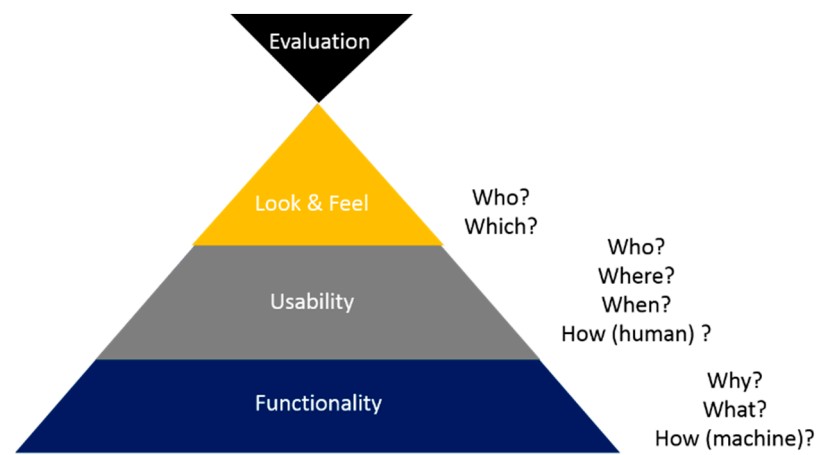

**Figure 2.** The FULE methodology is built like a pyramid, meaning there is a clear hierarchy.

The functionality phase focus on the technology part of the system, addressing a specific product's motive—"*Why* is this product needed?", offering a technical solution—"*What* is the suggested solution?", "*How* does it work?". After establishing the foundation and setting an initial framework considering all technical and functional requirements, it is time to proceed to the usability phase, which is all about the users—"*Who* is the intended user?", "*Where* and *when* will this product be in use?", and what is the intended user–product interaction—meaning: "*How* is it going to be used?". Having a primary concept for the intended device, the designer is able to proceed to the next phase: the look-and-feel of the product. This phase is again user-centered, hence deals with "*Who* is the user?", "*Which* design language would fit?". The final phase consists of an evaluation of the process as a whole.

### 5.1. Phase 1: Functionality

The functionality phase deals with the system itself. It probes the reason for this specific product and offers a technical solution. The designer's role and responsibility here are to learn and understand the functionality and requirements of a system. In some cases, the designer's work may affect the functionality, as users' needs are met during the research process.

This phase is divided into three sub-phases (two learning sub-phases and a design sub-phase): Background—understanding the motive for developing this new product (What kind of problem does this system address? What is the novelty of this product?). Requirements—analyzing system requirements and conflicting demands (electronic requirements, standards, materials (for example, strong adhesives, resistant materials for specific conditions, bio-compatibles, etc.)). Laying the Foundations—designing the initial framework. Figure 3 provides more details. The foundation phase can also be defined as the problem-solving phase in which the designer elaborates on the requirements and decomposes the primary function into practicable sub-functions; this step can be repeated for each sub-problem [17].

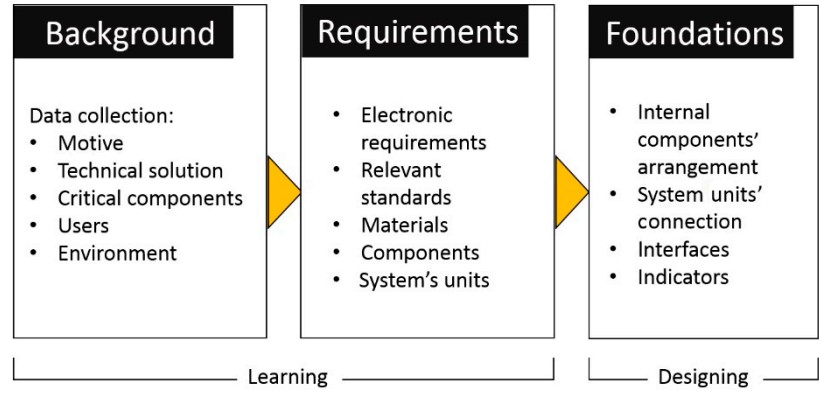

**Figure 3.** The functionality phase is divided into three sub-phases.

### 5.2. Phase 2: Usability

During the development process of a new medical device, attention is primarily focused on specifications that guarantee proper functionality. However, technical and functional specifications usually do not fully and thoroughly consider user needs or guarantee easy and safe usage [18]. Thus, technological products, even innovative and cutting-edge ones, might fail in the market when the design does not match human needs, cognitive processes, or the work environment [19]. The second phase, usability, refers to product features that would help users achieve their goals effectively and efficiently (ISO 9241-11). This phase is divided into four sub-phases: learning, analyzing, defining, and ideating. Here, significance in the product development process (and especially in the design of medical devices) cannot be underestimated. Poor usability may result in high costs, wasted

time, increased need for service calls and training, and decrease staff morale and productivity [20].

The understanding that design can prevent the user from making mistakes (or at least minimize them) is relatively new. In the past, it was customary to resolve usage faults by adding a warning to the operator's manual or a warning sticker on the product itself. Over time, it was recognized that if a trained user (a nurse or doctor) makes a mistake with a device, there is likely a design problem rather than a user problem.

Walking through hospital corridors can illustrate the importance of the development process' usability stage and reveal various pieces of evidence of events resulting from misuse; each product warning sticker tells a story. This understanding also influenced the Food and Drug Administration (FDA) regulators to emphasize human factors and usability and add tests for this before giving new systems approval. In 1997, the first FDA manual addressing medical device development professionals, *Do it by design: An Introduction to Human Factors in Medical Devices* [6], was published. The booklet provides a concise explanation of basic concepts such as physical and sensual characteristics, perceptual and cognitive abilities, and mental models. In addition, the booklet contained case descriptions from the Medical Device Reporting System [21], a system into which details of events that ended in injury to the body or mind are entered. According to the methodology proposed by the FDA, in an ideal situation, hardware and software designers, engineers, human factors personnel, document writers, and clinical staff coordinate their efforts to achieve a user interface design that ensures speed, convenience, and safety at every stage of a product's life: assembly, installation, use, and maintenance.

The usability phase of the FULE methodology uses the main principles described by the FDA. The phase consists of four sub-stages: the first two about learning and analysis, and the other two about settings, planning, and creating, as illustrated in Figure 4.

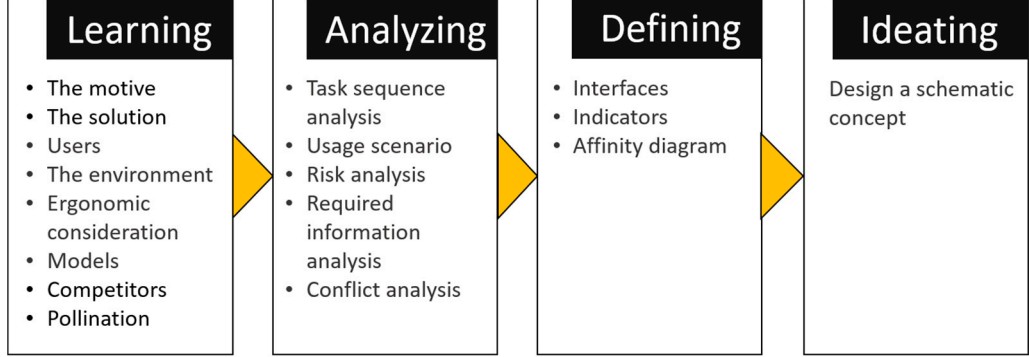

**Figure 4.** Usability phase.

The first stage is *learning*. The designer must clearly understand the new design's goal and become familiar with the medical condition and the offered solution. In this stage, the designer defines all users (patient, specialist, nurse, family, informal caregivers, etc.) and understands the environmental conditions (physical environment, social environment, and ambient conditions). The tools we use for the learning stage are varied and may differ from project to project. Still, they will always include examining an existing/similar system (if any), interviews with the various users, and observations in the work environment and similar procedures. Additional tools may be anonymous questionnaires, pollination (searching for ideas in tangential or other content worlds), a literature review, and consulting with experts from various fields.

After collecting the data, we perform the *analysis* phase by building task sequence models and usage scenarios. These models are used for risk analysis and required information and interface analysis (depending on system requirements and capabilities, and user needs), along with conflict analysis. The methodology uses graphical and visual tools to represent different product features and requirements.

Sometimes at this stage, it is found that some requirements conflict. The FULE methodology encourages decision making using a visual representation of the alternatives using the Harris profile [22] (our case study illustrates the use of this tool in Section 6.2) or by creating different graphs presenting the contradiction. Figure 5 illustrates an example of a diagram showing two conflict demands in a hand-held operational tool. In this case, the surgeons' standard tool was not ergonomic. The conflict was to design a more ergonomic device, replacing the usual holding method with a new one, or avoiding the change, preferring mental models and familiarity over ergonomics. Using the diagram helps the team communicate and understand the alternatives.

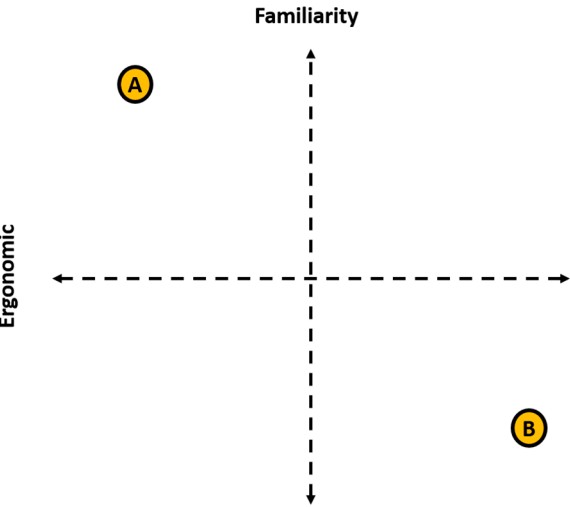

**Figure 5.** An example of a diagram showing two conflicting demands in a hand-held operational tool.

Upon completion of the usability phase, the designer must clearly define all interfaces and indicators in the defining stage. The result of the usability phase is ideating. Based on an affinity diagram determining all external features' optimal positions, the designer formulates an initial schema that considers all functional and usability requirements.

*5.3. Phase 3: Look-and-Feel*

By having a primary concept for the intended device, meeting all the functional requirements, leading the user towards the correct usage by placing all interfaces and indicators in their optimal location, and considering all usability requirements, the designer can proceed to the next phase: the look-and-feel of the product. Look-and-feel refers to the product's aesthetics and visual qualities such as colors, shapes, proportions, materials, etc., and the way intended users perceive them. The look-and-feel may express a very innovative product or, on the other hand, a familiar old friend. This phase is all about the structure, color, and materials chosen for the final product.

The product aesthetics and its look-and-feel are strategic tools used to gain a sustainable competitive advantage [23–25]. It is widely accepted that the designer has an important role in a product's success [26,27]. An aesthetic experience can trigger an emotional experience [28]. Aesthetic features in product form were found to be linked with a user's perceptions and the desire to own the product [29]. Designing for desirability, particularly when dealing with consumer goods, is important for achieving a positive reaction and a good user experience. The aesthetic criteria are found to be context and experience-dependent [30], meaning the designer must fit the design language to the intended users and domains. Contemporary products must be desirable; they should reach their user on many levels: emotional, social, and intuitive. While functionality and usability are the product's foundations, it will not succeed in the marketplace unless the product is also desirable.

In the FULE model, we replaced the term "desirability" with the term "look-and-feel," believing that the term "desirable" is not suitable for every product, particularly for medical devices. "look-and-feel" refers to the visibility of a product and how it is perceived by users, allowing a wider variety of feelings and perceptions such as innovative, credible, user-friendly, etc.

The purpose of this phase is, of course, to improve marketing and sales by creating an advantage in terms of the user's experience. However, furthermore, this phase's importance is in creating a long-term trust relationship with the user, along with the usage, acknowledging that trust is a key factor for safe usage.

Although the aesthetic design phase is often perceived as the more intuitive and creative stage, some may try to rationalize it in an effort to substantiate the design process. Benedek and Miner [7] developed a method to check the emotional response and desirability of a design or product called "Microsoft reaction cards." They created a desirability Toolkit, using 118 cards with different product-reaction words and phrases written on each one representing a broad spectrum of options and dimensions of desirability (for example, accessible, annoying, boring, exciting, familiar, etc.). This tool could help categorize varying designs or user groups (by gender, age, etc.).

To attach more quantitative values to emotional elements, we created three inspiration boards for medical devices. Each board contains pictures of devices with common visual characteristics. In an on-going process, using various focus groups (medical professional teams, marketing directors, design students, etc.), adjectives were attached to each board to depict potential users' perceptions, emotions, and reactions to the design characteristics, as shown in Figure 6. Participants were asked to ascribe descriptive words based on their first impression of the visual qualities without any further information regarding their functionality and context.

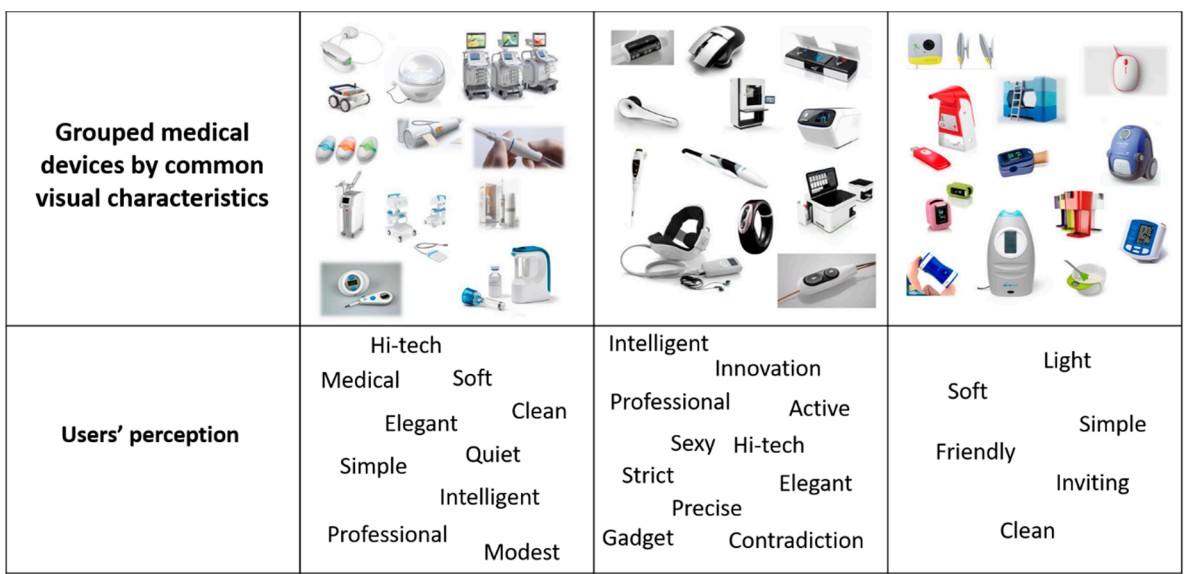

**Figure 6.** Medical device design inspiration boards and ascribed perception words.

Using these boards, we created three abstract diagrams to define guidelines for designing medical devices according to the desired user experience (marketing demands, or potential user expectations collected by interviews or surveys), as shown in Figure 7.

Using visual representations provides a designer with the means to solve conflicting demands and also to define precise specifications in the field of the product's appearance. It helps the designer communicate with different stakeholders regarding the preferred look and feel. Figure 8 shows an example of a graph presenting the appearance spectrum for the design of medical devices for children.

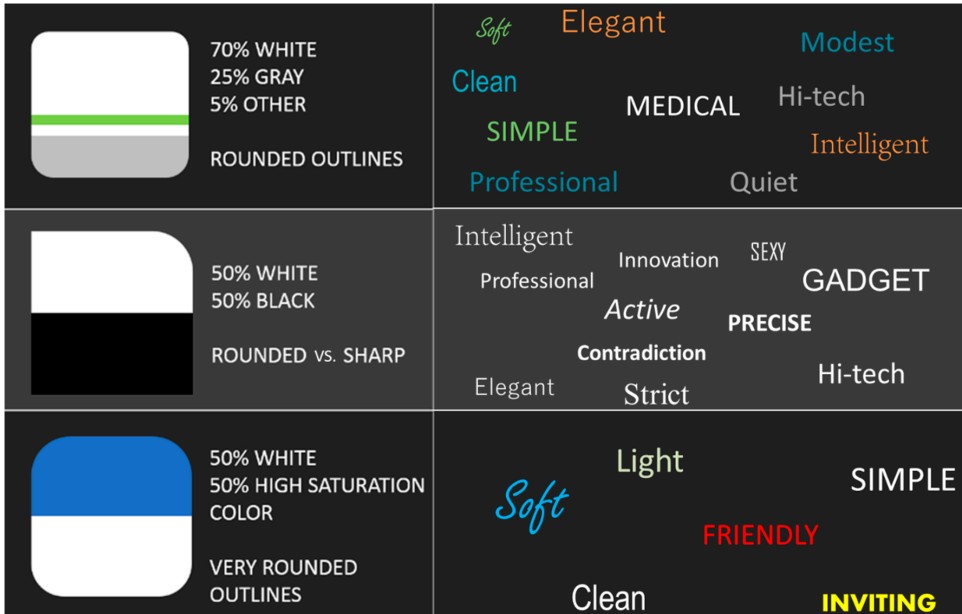

**Figure 7.** Abstract diagrams created to define guidelines for designing medical devices according to the desired user experience.

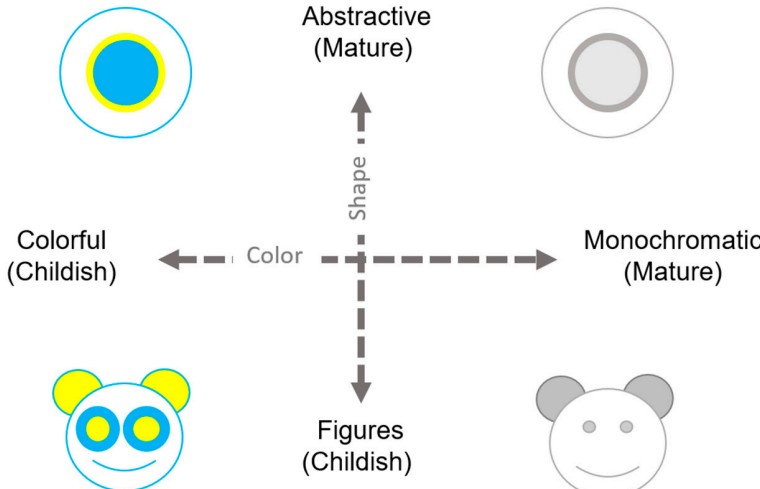

**Figure 8.** A graph presenting the appearance spectrum for the design of medical devices for children.

FULE encourages the designer to manage expectations by placing the future design on a chart defining its values of different visual qualities. FULE suggests using surveys and focus groups using intended users to complete the look-and-feel phase. These tools help the designer choose between design concepts and to evaluate the outcome.

### 5.4. Phase 4: Evaluation

This phase refers to a summative assessment of the system within its natural environment while evaluating whether the final product meets the criteria and expectations set at the beginning. In this stage, the product's three basic features are examined: Functionality—Does the product perform well? Usability—Is it convenient and clear for users? Does it allow a smooth and error-free workflow? Look-and-Feel—Does it address the target users? Does the product express the desired message?

Furthermore, this step may help in decision making about how to embed the product in terms of both planning and marketing. The former relates to user training solutions and the latter to defining sales and distribution routes (e.g., renting the system at a low-cost

price and charging a high price for designated disposable components, compared to a high first payment on the system and delivering low-cost components). The tools used to perform this phase include questionnaires, interviews, and observations at the time of use by the intended user. Figure 9 illustrates the summative assessment phase.

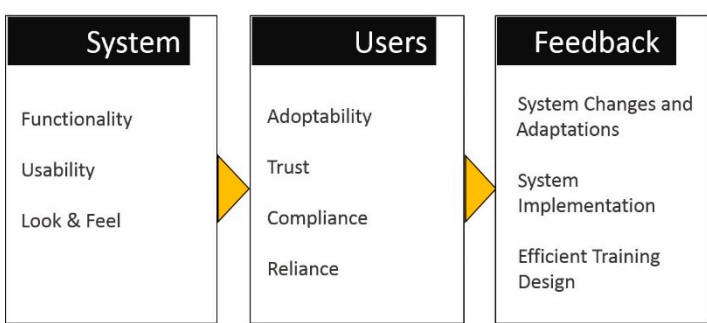

**Figure 9.** Summative assessment.

## 6. Case Study

To illustrate the advantages of the FULE methodology, we present a case study of the design process of a complementary autonomic breathing system by Hospitech®. The system aims to prevent ventilator-associated pneumonia (VAP) in respiratory patients. The design process was initiated even though the company already had a first functional prototype aiming to evaluate the reduction in VAP occurrences. Usability issues that were not considered during the first prototype development process resulted in operational challenges in hospitals, and nursing staff refused to handle the system due to its complexity. Hence, this project's goal was to improve its usability, along with achieving acceptance and trust. We used FULE in the development of the second generation of the product in an effort to improve user interaction. After operation by the nursing staff at (Rambam Medical Center, Haifa, Israel) and following the FULE methodology, the product was evaluated for its usability, look-and-feel, and trust.

### 6.1. Case study—Functionality

6.1.1. The Motivation

The need for mechanical ventilation is the primary reason for admission to intensive care units. Critical aspects in managing mechanically ventilated patients are related to unsuitable cuff pressure of the endotracheal tube. Underinflation of the cuff may lead to air leakage and improper ventilation. Consequently, contaminated subglottic secretions might aspirate into the lungs, possibly leading to ventilator-associated pneumonia (VAP) [31]. Besides being a significant cause of morbidity and mortality, VAP is known as one of the healthcare-associated infections (HAI) which burden the health system and can be preventable. Several studies have attempted to estimate VAP's economic burden, reporting costs ranging from USD 10,000 to USD 40,000 per treated patient [32]. On the other hand, the cuff's over-inflation might cause local mechanical injuries such as edema and ulcerations due to excessive pressure on the tracheal tissue.

6.1.2. The Solution

The AnapnoGuard system prevents these complications by integrating two solutions: First, using automatic inflation and deflation of the cuff pressure based on continuous monitoring of $CO_2$ leaks above the cuff, indicating air leakage from the lungs. When the system detects $CO_2$, it inflates the cuff, and when a leak is not detected, the cuff gradually deflates. This ensures tracheal sealing at minimal cuff pressure. The second solution is simultaneous control rinsing and evacuation of subglottic secretions from above the cuff.

### 6.2. Case Study—Literature Review

Our case study involves a medical device that uses autonomous features. These are becoming an essential part of the healthcare work environment. These features will convey a new means of human–technology interaction [33]. To identify and understand the components required for a successful user-centered product design process of an autonomous medical device, we must first define the concept of autonomous systems and human–autonomy teaming (HAT) and familiarize ourselves with the concept of user trust.

#### 6.2.1. Autonomous Systems

Automation is slowly becoming part of nearly every aspect of our lives and plays an increasingly role in everyday life, including in our home and workplace. We can already see the influence in some sectors being replaced by automated systems, such as bank tellers, cashiers, and gradually drivers and deliveries. According to some studies, in the upcoming decades, 47% of the currently existing occupations in America are at high risk of decline [34]. One of the evolving sectors is the healthcare workforce. Studies have shown that automation will not simply replace the human part, but rather change its role [35]. New roles will be required for the control and supervision of autonomous systems, and contemporary structures of co-workers will be created to operate them: human–automation teaming (HAT). As with every team working together, trust between members is essential to collective functioning.

Automation can be defined as technology that actively selects data, transforms information, makes decisions, or controls processes [36]. Automation implemented properly could increase efficiency, improve safety, lower operator workload, etc. [35]. Automation is a way of operating or controlling a process by automatic means. It refers to the full or partial replacement of a job that used to be completed by humans. Automation is not all or nothing but rather varies across a range of levels, from full manual performance to full automation. It is customary to use a scale of ten levels of automation (LOA) defined by Sheridan and Verplank [37]. The scale divides the levels of automation from full control of the operator (Level 1) to full control of the system (Level 10).

#### 6.2.2. User Trust

Establishing applicable trust in automation is an important factor for improving human–automation teams' safety and productivity [38]. Although automation manages most of the algorithmically intense workload within a socio–technical work system, the final decision-maker is the human operator. Hence, for a productive result, the human must accept the automation's output [39]. Factors affecting user trust toward the system can be divided into three categories: factors related to the operator, to the environment, and to the automated system [38].

#### 6.2.3. Summary

Our case study literature overview pointed out the importance of achieving users' trust when designing autonomous medical systems. We used that insight to set our design goals for this device: the future device must be functional, easy-to-use, trustworthy, safe, and user-friendly.

### 6.3. Case study—Usability

After laying the foundation and setting an initial framework considering all technical and functional requirements, it is time to proceed to the usability study. Our case study's design process began while the company already had a first functional prototype, but usability issues were not considered during the development process. FULE was used to develop the second generation of the product to improve user interaction with it. We started this phase by examining the functional prototype in search of possible failures and risks. Failures were encoded on a scale between 1–4, where the lowest rank is 1 = user frustration, and the highest is 4 = physical or mental injury.

Among the findings were deficiencies in both the design of the system itself and of its disposable units. The buttons were all aligned, and were the same size, shape, and color, with no hierarchy, and without distinguishing between urgent and non-urgent actions. In addition, the connectors were placed without considering the wiring direction, linking the various lines, and tangling them.

In the next step, we conducted observations at Rambam Hospital, monitoring the system in action and at two other hospitals (Assaf Harofeh, Be'er Ya'akov, Israel and Rabin Medical Center, Petah Tikva, Israel) watching nurses' regular work where there was no similar device. In addition, we interviewed the nurses and other medical staff to understand their needs, expectations, and requirements for the future product.

The next stage was creating usage scenarios for each possible human–technology interaction. Every scenario included all possible failures that might occur due to inappropriate usage. These usage scenarios indicated that the system's dimensions and weight made its mobility in the hospital almost impossible, and it often was in the staff's way. Nurses claimed it was often difficult to cope with the complex assembly of the system (before connecting it to a new patient), as it required gathering various components located in different rooms that were sometimes unavailable when needed. The assembling process itself appeared to be confusing for the staff, potentially causing user mistakes. It seems that at this stage, rather than making the task easier, the system was a burden for the staff.

Regarding the required information, all nurses agreed that the system should display only the most essential information (cuff pressure and critical alarms). Observations and interviews' outcomes were used to help us design three different concepts of the new device. The top three suggestions for the system location were: (1) attached to patient's bed, (2) connected to the back wall, and (3) using a designated pole. We used Harris's profile [22] to help us visualize each design concepts' strengths and weaknesses according to our declared requirements (Figure 10).

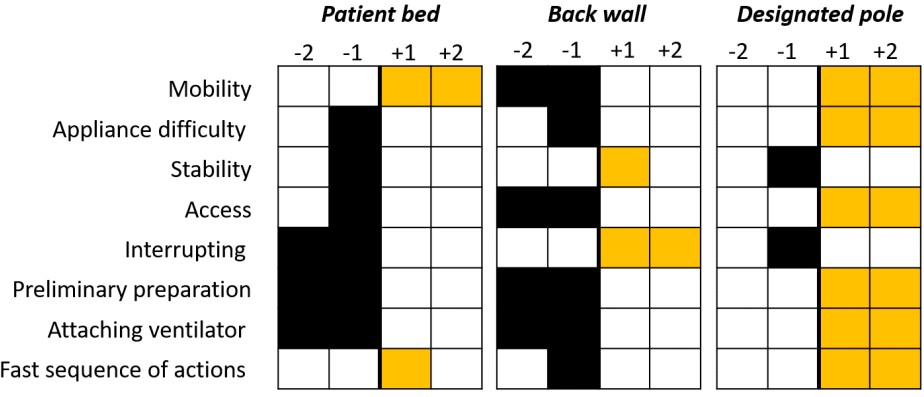

**Figure 10.** Using the Harris profile for concept comparison in our case study.

Using this four-scale scoring with a graphic representation helped us easily compare between our three options (derived from conducted interviews and observations) and we chose to design a designated pole that fit the design requirements in the best way.

### 6.4. Case Study—Look-and-Feel

In a discussion conducted with the nursing staff from Rambam Medical Center, who participated in the functional system's clinical trial, the guidelines for the visibility of the product were defined. It was decided that the system should look, first and foremost, friendly, and inviting; it should also look simple, easy to use, and credible. They sought to avoid innovative and futuristic designs and preferred one that would easily integrate with other devices. Hence, the chosen outlines were very simple, slightly rounded, with a white and blue color scheme, representing calmness and quiet.

### 6.5. The Final Product

Our research indicated that the main problem of the functional system lay in its complexity. Usage scenarios of both a new patient connection and the preparation of the next patient system were time-consuming, complex, and prone to user mistakes and failures. It was decided to replace the entire mass of tubes with one disposable connection kit (cassette) containing all the tubes according to the correct exit direction.

A large screen in the front of the product accompanies the user in the connection stages and preparation and provides information throughout the device's operation. A relatively large screen selection stemmed from the need to display the cuff pressure prominently so that it could be seen anywhere in the room. We chose to locate the screen on a large door that covers the entire front of the product. The purpose is to hide the cassette and tubes so that the product visually looks clean and pleasing to the eye.

The only remaining key at the front of the product is the standby key. The remaining non-urgent keys were moved to the back screens of the touch screen. The power button is located below the screen and is sunken. Besides the screen that provides the system activity information, two additional LEDs were added to indicate the device status. Figure 11 presents the AG100s final design.

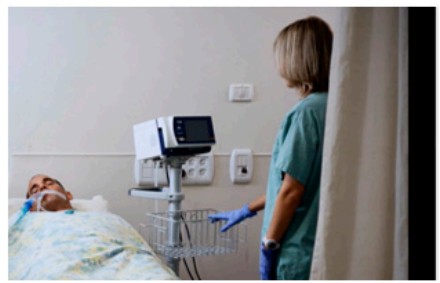
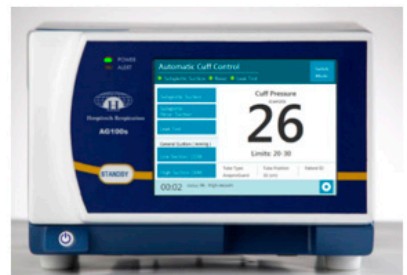

The system in its natural environment | Control unit

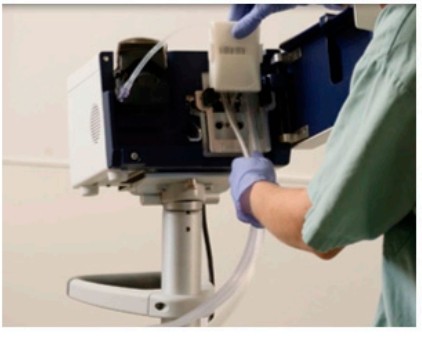
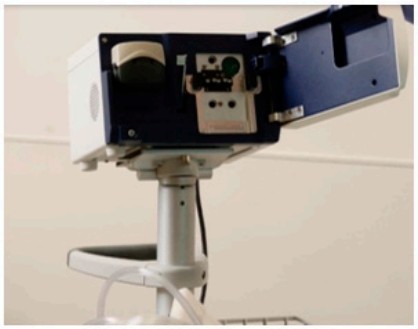

Inserting disposable connection kit | Open door system

**Figure 11.** AG100s final design.

The system contains, besides the control unit (AG100s), two additional disposable components: the AnapnoGuard endotracheal tube and a disposable connection kit, as shown in Figure 12.

### 6.6. Case Study—Evaluation

This stage aimed to evaluate our final design for the AG100s system, based on the goals and expectations set at the beginning of the design process. Did we succeed in creating a user-friendly device perceived as trustworthy? Had we gained the intended users' trust and acceptance?

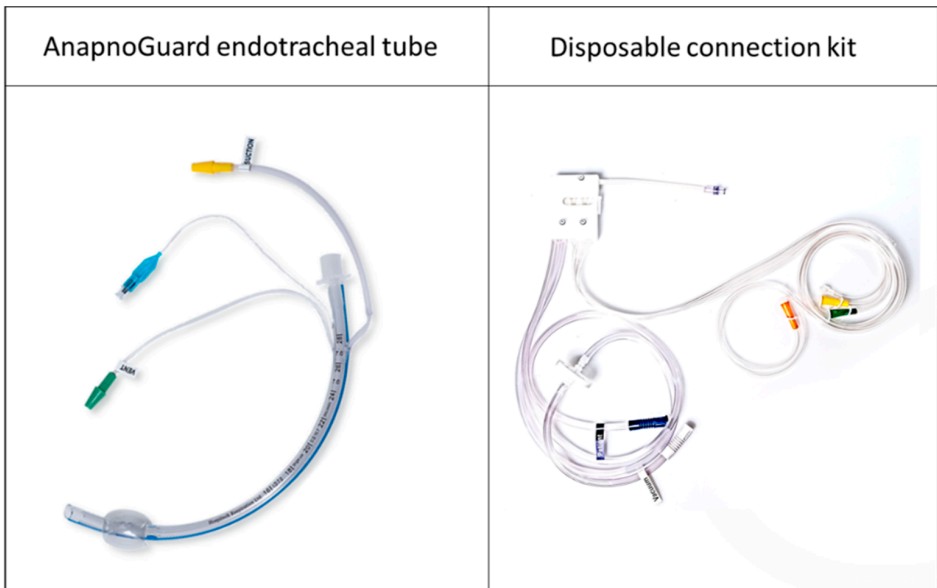

**Figure 12.** Single-use disposables.

### 6.7. Data Gathering

The evaluation phase was conducted at the ICU cardiac surgery units at Rambam Medical Center among trained nursing staff who were using the new AnapnoGuard system for six months and at (Kaplan Medical Center, Rehovot, Israel) where there is no similar autonomic system. Hence, the team there must hand operate the actions that the AG100s system performs automatically.

Phase one: exploratory interviews. Six semi-structured interviews were conducted. Participants included six volunteering nurses, three males and three females who had experienced the AG100s system. The interviews took place at Rambam Medical Center. The participants were asked to share their knowledge, opinions, and perceptions regarding ventilator-associated pneumonia (VAP), the AG100s system's usability and appearance, and autonomic medical systems in general. All interviews were recorded and transcribed in full. The data were analyzed thematically.

Phase two: closed-ended questionnaires. This stage was conducted in two different groups: Group A: nursing staff who were using the AG100s system at Rambam Medical Center, and Group B: nursing staff who were not familiar with the AG100s system at Kaplan Medical Center. The second group had a short training process that was similar to the routine training procedure provided at the introduction of the AG100s system at Rambam the first time. Following training, participants were asked to complete the questionnaires based on their impressions. The questionnaire included closed-ended questions according to the following chapters: Chapter A—VAP, Chapter B—Technology, and Chapter C—The AG100s system.

The data collection process involved 14 questionnaires answered by medical staff who had experienced using the system at the Rambam Medical Center, two of which were eliminated because they were not fully completed. The remainder included 11 questionnaires filled out by the nursing staff and one completed by a physician. Thus, the results address only the 11 questionnaires completed by the nursing staff. Table 1 presents participant data from Rambam Hospital.

In addition, we gathered 22 questionnaires from members of staff who did not experience the system themselves but who underwent initial training at the Kaplan Medical Center. Because most of them chose to ignore the question of seniority, the analysis did not address this category. Table 2 presents participant data from Kaplan Medical Center.

**Table 1.** Participant data from Rambam Hospital.

|  |  | N |
|---|---|---|
| Total |  | 11 |
| Gender | Males | 6 |
|  | Females | 5 |
| Age | <35 | 3 |
|  | 36–45 | 6 |
|  | 46–55 | 1 |
|  | 56< | 1 |
| Seniority (years) | <10 | 3 |
|  | 11–20 | 6 |
|  | 21–30 | 1 |
|  | 31–40 | 1 |

**Table 2.** Participant data from Kaplan Medical Center.

|  |  | N |
|---|---|---|
| Total |  | 22 |
| Gender | Males | 7 |
|  | Females | 15 |
| Age | <35 | 8 |
|  | 36–45 | 4 |
|  | 46–55 | 6 |
|  | 56< | 4 |

Participants filled out two personal questionnaires regarding their attitude toward technology and their tendency to adopt new technologies. They were asked to rank their frequency of use of various technologies such as GPS and cloud backup on a scale of 0 = never to 5 = almost daily. The results suggest a slight gender difference, as women responded that they used a wider variety of technologies than men. The age factor did not impact the use of technology. In both hospitals, men attested to a slightly more positive attitude toward technology than women.

Evaluating the correlation between the use of technology and attitude toward technology did not reveal any significant results. Participants attesting to technological orientation did not necessarily employ new technologies, and participants using technologies were not necessarily technology proficient.

To evaluate the system's aesthetic perception, the participants were asked to rank their impression of its visual appearance and graphical user interface (GUI). We used adjectives such as friendly, innovative, pleasing to the eye, etc. Women gave the system's look-and-feel a higher score than men at both hospitals. Participants rated the visual appearance as pleasing to the eye (4.54 out of 5), medical (3.9 out of 5), and reliable (4 out of 5), with no significant differences between male and female responses except for device friendliness. Female nurses perceived the design of the system as friendlier (4 out of 5) than male nurses (3.16 out of 5) (Figure 13).

To evaluate usability, we used a standard system usability scale (SUS) [40] aiming to examine users' impression of the comfort of use of the system in general, along with a designated questionnaire asking the respondents to describe the degree of comfort and clarity of use for each of the system's operating phases. It was found that female nurses rated the comfort and clarity of use slightly higher than male nurses, and the scores given were mostly high (above four). Results also suggest that the older the respondent, the higher the ranking. However, the system's SUS score was high on average (64.32 when the minimum required is 68). Grades ranged from 45 to 87.5 and indicate that there is a clear gender distinction. Male nurses gave higher grades in general (males: 70, females: 57.5). We also found an inverse ratio between the grades and age factor: the older the respondent, the lower the grade—Table 3 sums up the data from both questionnaires.

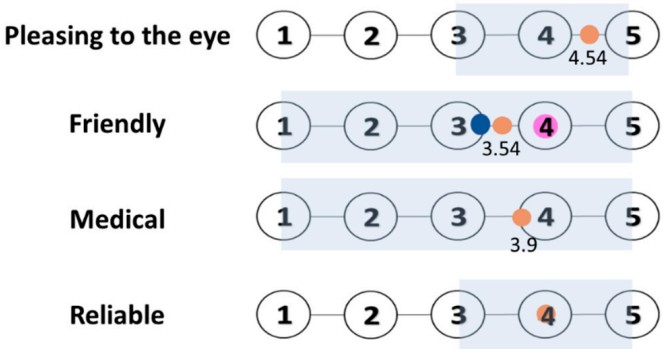

**Figure 13.** Evaluating the aesthetic perception of the system.

**Table 3.** Summary of the system usability scale (SUS) and designated usability questionnaire.

|  |  | N | Physical Usability (0–5) | SUS (0–100) |
|---|---|---|---|---|
| Total |  | 11 | **4.28** | 64.32 |
| Gender | Males | 6 | 4.17 | **70** |
|  | Females | 5 | 4.41 | **57.5** |
| Age | <35 | 3 | 4.15 | 76.67 |
|  | 36–45 | 6 | 4.23 | 62.92 |
|  | 46< | 2 | 4.65 | 50 |

The last part relates to the issue of trust in the system. The questionnaire included 24 positive and negative statements regarding general confidence in the system and trust in specific actions. Participants were asked to rank their level of agreement for each one.

In order to calculate the degree of trust, we considered only questions addressing general trust in the system, excluding questions relating directly to specific operations or concerning trust in general. At the same time, we calculated the average grade participants gave specific operations. Table 4 sums up these data.

**Table 4.** Summary of the trust questionnaire.

|  |  | N | Specific Feature Trust (0–5) | General Trust (0–5) |
|---|---|---|---|---|
| Total |  | 11 | 3.72 | 3.17 |
| Gender | Males | 5 | 3.88 | 3.45 |
|  | Females | 6 | 3.53 | 2.82 |
| Age | <35 | 3 | 4 | 3.5 |
|  | 36–45 | 6 | 3.5 | 3.04 |
|  | 46< | 2 | 4 | 3.06 |

Women expressed a lower level of trust toward the AnapnoGuard system in every tool used for the study. In addition, women expressed a lower level of trust toward technologies in general, yet indicated they adopt and use new technologies more often than men (Rambam) or in an equal manner (Kaplan). Figure 14 illustrates the correlation between gender and trust.

Among male nurses, we found a definite link between the system's SUS score and users' trust; a positive correlation exists (R = 0.752) between the SUS score and the trust level ($p < 0.05$), as shown in Figure 15. These findings match previous studies' conclusions that found a definite link between the two [16].

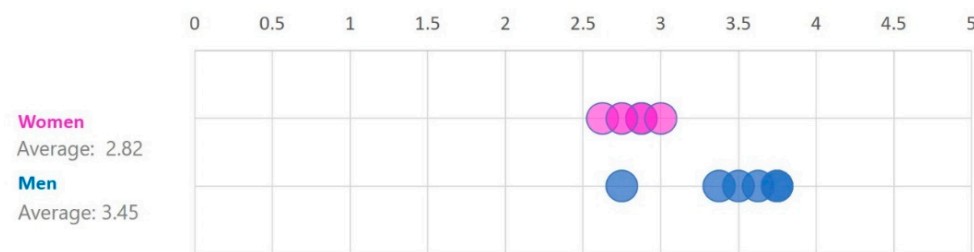

**Figure 14.** Correlation between gender and trust.

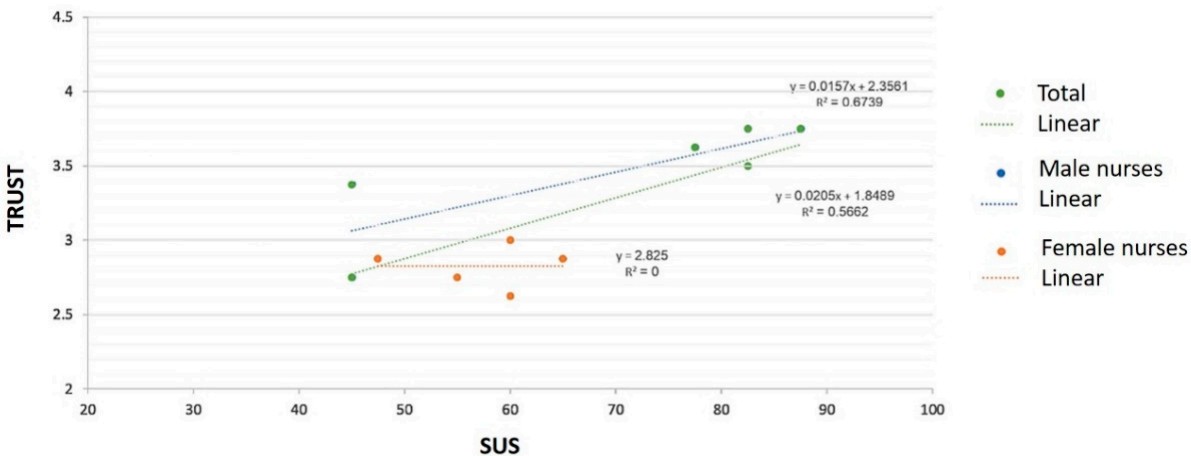

**Figure 15.** Correlation between SUS score and users' trust level.

The trust level expressed by men in both hospitals was almost identical. Experience with the system did not change their perception. On the other hand, among female nurses, we found differences between those who had experience with the system (Rambam) and those who only received the training (Kaplan). Kaplan's nurses expressed a higher level of trust.

It seems that the factors affecting the women's trust are different; some of these can be inferred from the interviews that we conducted, in which two main topics were raised. First is understanding the system mechanism. Participants were asked to explain how the AnapnoGuard system works. It seems that all the male nurses understood the system's rationale; all of them were familiar with the $CO_2$ sensor, as opposed to the female nurses who did not know how to explain what was happening. Perhaps this is the reason they found it difficult to trust the system. One of the men also pointed this out when he was asked if he trusted the system, saying: "At first, I was skeptical, but after understanding its rationale, I trust it." The level of trust is possibly related to the level of understanding. Another possible explanation for the different results between males and females is their perception of the human factor. In our interviews, we asked the participants if they could imagine a future where autonomous systems would replace human nurses. While all said no, their reasoning was very different.

Women spoke in terms of compassion and contact: "The patient needs to have someone to hold his hand." The word "communication" was repeated in all three interviews: "Without a doubt, patients need to communicate with a person." Among male nurses, the reasons were related to the human brain: "Nothing can compare with the human mind in understanding the overall picture"; "There is a need for a person to integrate all the information."

These results could help improving user trust by incorporating changes in the system itself, such as adding communication, written or verbal, or informing the user regarding its actions in real-time. Another adjustment could be to reach a higher level of transparency by explaining the reason for the system's actions and revealing the next move. Second, user trust could also be improved by designing a slightly different training program to make sure all users fully comprehend the way the system operates.

The nature of our case study, however, is subject to several limitations. The research tools utilized to evaluate the system are subjective; therefore, respondents' answers may be biased. In addition, due to staff turnover, we ended up with a relatively small sample size. This is a common situation due to the fact that we were evaluating a product for the first time with real users, employing a small group of participants.

## 7. Discussion

Analysis of our test case data suggests that the development process, according to the FULE methodology, generated a user-friendly product with a high grade of 4.28 (out of 5) on average. Participants ranked on average each of the usage step's *ease of use* with a rating of four and above, except for patient details uploading (which addresses the GUI of the product and not physical interaction with the system itself). Interviews also confirmed that the system is clear and comfortable for use. In one case, a participant who had previously worked with the former prototype even noted that "the updated device was much more sympathetic."

In terms of look-and-feel, women gave a higher score to the system's general look, with a grade of 3.95 as opposed to the men, who graded it as 3.67 on average. For the statement: "The system is pleasant to the eye," participants at Rambam Hospital rated it on average at 4.54, while participants at Kaplan Hospital rated it as 4. In the course of the look-and-feel stage, we defined the desired look of the system as friendly. In addition, it needed to come across as easy to use, integrating into the surroundings, and reflecting credibility. Analysis of the statements in detail indicates that women ranked the system as "seemingly user friendly" with a high score of 4 out of 5, whereas men rated it as 3.16. For the statement: "The system comes across as easy and clear to use," participants rated it on average as 3.73, with no significant difference between men and women. For the statement: "The system blends in with the surroundings," participants rated it on average as 3.82, with women scoring it higher than men. For the statement: "The system relays credibility," participants rated it as four on average, with men ranking it higher than women.

Results indicate that, for certain statements, there was an agreement between men and women; however, the statement: "The system comes across as friendly" generated different gender-related outcomes. The reason for this could be the relatively limited sample size. Another explanation may be that men and women perceive the term "user friendly" and its manifestation in a product differently. However, it should be noted that the Kaplan Hospital results did not reflect this difference, as ratings to all statements showed no gender differences except for "The system comes across as a medical device." Women rated this statement as 4.46, whereas men scored it as 3.6 on average.

## 8. Conclusions

In this study, we sought to develop a novel design methodology referred to as FULE in an effort to facilitate an educated and efficient bias-free design process of complex medical devices. We assume that working according to the FULE guidelines could lead to the design of better solutions for design challenges. The new methodology addresses various aspects of the product: functionality, usability, and look-and-feel while considering safety and user experience. The case study illustrates how the use of the FULE methodology provides the designer with tools to better select among design alternatives and reduce bias and subjective decisions in the design of an autonomous medical device.

Our case study addressed transformations in our surroundings, some of which necessitate integrating autonomous systems into various fields, including the medical realm.

The realization that autonomous systems are increasingly penetrating into our daily lives precedes the development of trustworthy systems to enhance safety and efficiency. Incorporating these systems and creating new structures in the form of human–autonomy teaming induces the recognition of trust as a key element when developing a system. Consequently, trust should be considered as one of the primary goals when developing a new autonomous system. The issue of acquiring user trust was widely explored throughout our test case. This stage's purpose was to fully comprehend and isolate the factors that could improve user trust, specifically in relation to autonomous medical systems.

In addition, we tested the usability of the system and matched it visually to our target audience. The general appearance of the product was rated as "pleasant to the eye". We found that the exterior design of the system creates the desired experience as defined by the users.

Analysis of the data from our test case indicates that the design's development, according to the FULE methodology, yielded a functional, trustworthy, safe, user-friendly product. Compared to the original device that was not developed using FULE, the development of the second generation determined that the FULE methodology applied in this case contributed to the device's success. Nevertheless, this does not guarantee that applying the methodology in a different case will also succeed. Future research should study these aspects.

Our data also indicate that the primary factor impacting the level of trust is gender. This was reflected in answers to both questionnaires distributed among hospital staff, as in the interviews. The interviews suggest that women find it more difficult to understand a system's rationale, possibly undermining their trust level in the device. Future research could be conducted to verify this assumption and suggest an alternative design experience to support it.

Having recognized that trust is essential to operate autonomous systems efficiently, we conclude that it should be acknowledged as a key element in the initial stages of the design process. Future studies may explore further adaptations to the FULE methodology to further promote trust. Additional research is required to adapt FULE to additional complicated systems (outside of the medical field), necessitating user-centered design and user trust.

**Author Contributions:** Conceptualization, E.L.-P. and Y.B.; methodology, E.L.-P. and Y.B.; validation, E.L.-P. and Y.B.; formal analysis, E.L.-P.; investigation, E.L.-P. and Y.B.; resources, E.L.-P.; data curation, E.L.-P.; writing—original draft preparation, E.L.-P.; writing—review and editing, Y.B.; visualization, E.L.-P.; supervision, Y.B.; project administration, E.L.-P. and Y.B. All authors have read and agreed to the published version of the manuscript.

**Funding:** This research received no external funding.

**Institutional Review Board Statement:** The study was approved by the Ethics Committee of Ben-Gurion University of the Negev (18-October-2018).

**Informed Consent Statement:** Informed consent was obtained from all subjects involved in the study.

**Data Availability Statement:** The data presented in this study are available on request from the corresponding author. The data are not publicly available due to ethical restrictions.

**Conflicts of Interest:** The authors declare no conflict of interest.

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
