# Peer review of "FULE—Functionality, Usability, Look-and-Feel and Evaluation Novel User-Centered Product Design Methodology—Illustrated in the Case of an Autonomous Medical Device"

_applsci, doi:10.3390/app11030985_

Round 1
Reviewer 1 Report
The paper presents FULE, a design methodology aimed at supporting decision-making in the various stages of the product development. FULE acts as a container of methods and tools, that the authors systematised throughout their direct experience and involvement in design projects. In particular, the authors discuss their proposition by reporting the application of the framework in the scope of healthcare design.
Although the authors certainly tackles relevant design themes, from the understanding of the design process, to the development of appropriate approaches to the design of effective medical devices, the paper presents major flaws which make the manuscript hardly acceptable in its current state.
The contents are a rich source of insights that can potentially spark reflection and novel direction in design research, yet there are three orders of issues that severely affect the manuscript. I try to summarise in the following, and in particular:
1. Overall methodological research approach - Although the research is inherently project-based, its management and evaluation are not supported by evidence, beyond the general statement that the project outcome was successful (and I believe it was, the outcome is impressive). This is not to say that design research should strictly adhere to the scientific method, yet the outcome should be evidence-based. The impression is that the authors made a special effort in self-assessing their design action in order to generalise a successful process into a design methodology.
In particular the authors state that FULE contributes to reducing (cognitive) bias and subjective decisions. This statement is problematic for at least two reasons. First, no discussion nor evidence is provided that FULE actually reduces cognitive bias.
As a reader, I would have expected at least some pointers to protocol analysis approaches to design cognition. See e.g.
* Hallihan, G. M., Cheong, H., & Shu, L. H. (2012, August). Confirmation and cognitive bias in design cognition. In International Design Engineering Technical Conferences and Computers and Information in Engineering Conference (Vol. 45066, pp. 913-924). American Society of Mechanical Engineers.
* Gero, J. S. (2011). Fixation and commitment while designing and its measurement. The Journal of Creative Behavior, 45(2), 108-115.
Second, it is not clear how FULE contributes to objective decision-making, as the tools and methods briefly reported throughout the text mostly refer to qualitative inquiries (survey and observations). It is quite significant the high rate of expressions like "results suggest, this could help" spread throughout the text.
2. Contribution to the field - As a reader, I was very curious to gain insights and learn from this project-based research. Yet, I was confronted with a very descriptive approach, which unfortunately fails to convey or make explicit the contribution in terms of knowledge. It feels like the authors did not find the right key to unfold the tacit knowledge that they developed during the project development. The overall message is left entangled. In the practice, the question is "how would designers and researchers benefit from the authors' experience, when dealing with similar cases?".
3. Scope of the contribution - The paper aims at introducing FULE as a design methodology to improve the product development. Yet, I would rather say that FULE is rather a framework, a systematic container of well-established tools and methods whose application has been optimised and tailored to the specific needs of the design context at hand (i.e., healthcare). Nonetheless, the discussion of the design case ends to focus on gender and age effect in the appraisal of the product, and only by the end of the paper the concept of trust is apparently discovered and introduced.
In this respect, I invite the authors to make the scope clearer, which I believe will flow easily once the methodological approach is revised. Finally, I well know that providing a convincing narrative capable to hold the big picture, while providing enough detailed reasoning and proof, is a complex task. And typically one makes use of references to previous works as pointers. Though, I could not find any previous work by the authors, in this respect it may be more effective to introduce FULE gradually, by publishing separate works that delve in more depth in the many aspects of this framework.
-----------
Comments
1. Introduction
This section feels quite vague and tautological. It does not clearly state what is the problem.
A description of the paper structure would help the reader to get familiar with the overall reasoning, and to navigate the contents.
- p2, l65: the visual means are not discussed, it is not clear in which way conflicts are resolved and communication improved.
2. Background
The section is quite divulgative, that is any designer well knows how to approach a project, from the analysis, to the ideation, through development and evaluation. It is not clear to which topic this section is background. Is it decision-making, design cognition, communication?
3. Aim
- p3, l119: the main ambiguity of the paper is condensed in this line. Is the paper about the project implementation or the discussion of the FULE framework?
4. Method
I would not call this section "Method" since there are no experimental data. Rather, this text offers a summary of the paper structure.
5. FULE methodology
In general, the whole section is very descriptive, as above-mentioned the main flaw is the lack of a proper evaluation of the framework.
- p4, l140: which are the several tool and methods used in FULE? How did you distill their optimal use?
- p4, l143: how do the four stages manage to support the role and responsibility of the designer? Which is exactly his role and responsibility according to FULE?
6- Case study
- p11, l392: Figure 10 -> figure 4
- p12, l395: Figure 4, how do you read that and which information should provide to the reader, in the overall scope of discussing the FULE framework?
- p15, l488: the authors discuss various gender and age effects related to the appraisal of the product. Beside the pure descriptive statistics approach, what do these results mean in terms of design opportunities and understanding?
The discussion of trust remains in general on a very speculative level. The authors claim the limitation of their study, and contradictorily state that the subjectivity of the evaluation approaches could have biased the results. But wasn't FULE aimed at supporting a more objective decision-making?
Author Response
Reviewer 1:
- Introduction
This section feels quite vague and tautological. It does not clearly state what is the problem.
A description of the paper structure would help the reader to get familiar with the overall reasoning, and to navigate the contents.
We thank the reviewer for this comment; we added a short description in section 1. Introduction:
p2 l68: "This paper presents an overview of the FULE methodology phases, followed by a full case study demonstration."
- p2, l65: the visual means are not discussed, it is not clear in which way conflicts are resolved and communication improved.
we thank the reviewers for their comment. We elaborated in section 5.2 usability and in section 5.3 look-and-feel about the different visual means. Section 6.2 case study- usability provides another example of using Harris profile for concept comparison in our case study
- Background
The section is quite divulgative, that is any designer well knows how to approach a project, from the analysis, to the ideation, through development and evaluation. It is not clear to which topic this section is background. Is it decision-making, design cognition, communication?
We thank the reviewer for this comment. We added a preface to the background section:
p2 l71 "In order to identify and understand the components needed for a successful user-centered product design process of a new medical device, first, we need to define medical devices and their users and understand the nature of their environment. We must also get familiar with other design methodologies and approaches and understand what the properties are of a design/development approach that should lead to the design of trustworthy products."
- Aim
- p3, l119: the main ambiguity of the paper is condensed in this line. Is the paper about the project implementation or the discussion of the FULE framework?
We thank the reviewer for this comment– we rephrased section 3. Aim:
p3 l126: "The primary objectiveof this research is to define a new medical equipment design methodology: FULE - Functionality, Usability, Look-and-Feel, Evaluation). FULE methodology was developed in order to help designers to design products with high usability and affordance. . By applying this methodology, the design team is guided to reason every decision made in the design process. Thus, decisions are not biased by the subjective opinion of the designer. In addition, the methodology provides a basis to resolve conflicting demands, and provides tools for discussion and alternative selection that leads to better product design.
In this paper, we demonstrate some of the FULE phases and tools in several design projects. In addition, we present a case study of the full design process of a complementary autonomic breathing system by Hospitech®."
- Method
I would not call this section "Method" since there are no experimental data. Rather, this text offers a summary of the paper structure.
We accept the reviewer's recommendation. We changed the title to: Study structure
- FULE methodology
In general, the whole section is very descriptive, as above-mentioned the main flaw is the lack of a proper evaluation of the framework.
- p4, l140: which are the several tool and methods used in FULE? How did you distill their optimal use?
- p4, l143: how do the four stages manage to support the role and responsibility of the designer? Which is exactly his role and responsibility according to FULE?
We thank the reviewer for these two comments. The information is provided for each phase in its section.
5.1 functionality:
p5 l186: "The foundation phase can also be defined as the problem-solving phase in which the designer elaborates on the requirements and decomposes the primary function into practicable sub-functions; this step can be repeated for each sub-problem [17]."
p5 l178: "The designer's role and responsibility here are to learn and understand the functionality and requirements of a system. In some cases, the designer's work may affect the functionality, as users' needs are met during the research process."
5.2 usability:
p6 l211: "This understanding also influenced FDA regulators to emphasize human factors and usability and add tests for this before giving new systems approval. In 1997, the first FDA manual addressing medical device development professionals, Do it by design: An Introduction to Human Factors in Medical Devices [6], was published……… The usability phase of the FULE methodology uses the main principles described by the FDA."
p7 l243: "FULE methodology encourages decision making using a visual representation of the alternatives using the Harris profile [21] (our case study demonstrates the use of this tool in Section 6.2) or by creating different graphs presenting the contradiction."
p6 l228: "The designer must clearly understand the new design's goal and become familiar with the medical condition and the offered solution. In this stage, the designer defines all users (patient, specialist, nurse, family, informal caregivers, etc.) and understands the environmental conditions (physical environment, social environment, and ambient conditions)."
In section 5.3 Look and Feel we describe how we use Benedek and Miner's Microsoft reaction cards in our methodology.
6- Case study
- p11, l392: Figure 10 -> figure 4
- p12, l395: Figure 4, how do you read that and which information should provide to the reader, in the overall scope of discussing the FULE framework?
We thank the reviewer for this comment. We elaborated as followed:
p12 l438: "We used Harris's profile [21] to help us visualize each design concepts' strengths, and weaknesses compare different concepts (Figure 10). "
p13 l443: "Using this four-scale scoring with a graphic representation helped us easily compare between our three options (derived from conducted interviews and observations) and chose to design a designated pole that fit the design requirements in the best way."
- p15, l488: the authors discuss various gender and age effects related to the appraisal of the product. Beside the pure descriptive statistics approach, what do these results mean in terms of design opportunities and understanding?
We thank the reviewer for this comment. In our case study's evaluation phase, gender was found to have a significant effect on users' trust. These results can influence different aspects of the system itself and its implementation, as detailed in section 6.7 data gathering:
p19 l594: "These results could help improving user trust by incorporating changes in the system itself, such as adding communication, written or verbal, or informing the user regarding its actions in real-time. Another adjustment could be to reach a higher level of transparency by explaining the reason for the system's actions and revealing the next move. Second, user trust could also be improved by designing a slightly different training program to make sure all users fully comprehend the way the system operates."
The discussion of trust remains in general on a very speculative level.
We thank the reviewer for this comment. We elaborated on the importance of trust in autonomous systems as presented in our case study by adding a new section: section 6.2 Case study- literature review:
p11 l373:
“Our case study involves a medical device that uses autonomous features. These are becoming an essential part of the healthcare work environment. These features will convey a new way of human–technology interaction (Bitan & Patterson, 2020). To identify and understand the components required for a successful user-centered product design process of an autonomous medical device, we must first define the concept of autonomous systems and human-autonomy teaming (HAT) and familiarize ourselves with the concept of user trust.
6.2.1 Autonomous systems
Automation is slowly becoming part of nearly every aspect of our lives and plays an increasingly great role in everyday life, including in our home and workplace. We can already see the influence in some sectors being replaced by automated systems, such as bank tellers, cashiers, and gradually drivers and deliveries. According to some studies, in the upcoming decades, 47% of currently existing occupations in America are at high risk of decline (Frey & Osborne, 2013). One of the evolving sectors is the healthcare workforce. Studies have shown that automation will not simply replace the human part, but rather change its role (Parasuraman & Manzey, 2010). New roles will be required for the control and supervision of autonomous systems, and contemporary structures of co-workers will be created to operate them: human-automation teaming (HAT). As with every team working together, trust between members is essential to function collectively.
Automation can be defined as technology that actively selects data, transforms information, makes decisions, or controls processes (Lee & See, 2004). Automation implemented properly could increase efficiency, improve safety, lower operator workload, etc. (Parasuraman & Manzey, 2010). C. E. Billings claimed automation must be human-centered. In his book, "Aviation automation: The search for a human-centered approach", he explains automation must ensure that the operator remains responsible for the safety and efficacy of the system. According to Billings, automated systems should never be allowed to operate or fail silently but must always inform the operator of their status.
Automation is a way of operating or controlling a process by automatic means. It refers to the full or partial replacement of a job that used to be completed by humans. Automation is not all or nothing but rather varies across a range of levels, from full manual performance to full automation. It is customary to use a scale of ten levels of automation (LOA) defined by Sheridan and Verplank (1978), The scale divides the levels of automation from full control of the operator (Level 1) to full control of the system (Level 10).
.
6.2.2 User Trust
Establishing applicable trust in automation is an important factor for improving human-automation teams' safety and productivity (Hoff & Bashir, 2015). Although automation manages most of the algorithmically intense workload within a socio-technical work system, the final decision-maker is the human operator. Hence, for a productive result, the human must accept the automation's output (Miller & Perkins, 2010).
Factors affecting user trust toward the system can be divided into three categories: factors related to the operator, to the environment, and to the automated system(Hoff & Bashir, 2015):
6.2.3 Summary
Our case study literature overview pointed out the importance of achieving users' trust when designing autonomous medical systems. We used that insight to set our design goals for this device: the future device must be functional, easy-to-use, trustworthy, safe, and user-friendly.
The authors claim the limitation of their study, and contradictorily state that the subjectivity of the evaluation approaches could have biased the results. But wasn't FULE aimed at supporting a more objective decision-making?
We thank the reviewer for this comment. The mentioned limitation refers only to the evaluation phase in our case study and appears in that section. As written:
p19 l600: "The nature of our case study, however, is subject to several limitations. The research tools utilized to evaluate the system are subjective; therefore, respondents' answers may be biased. In addition, due to staff turnover, we ended up with a relatively small sample size. This is a common situation due to the fact that we were evaluating a product for the first time with real users, employing a small group of participants."
Reviewer 2 Report
The authors present an interesting paper with a methodology for the development of functionally optimal products that guarantee a good user experience. Specifically, they do so by focusing on the design of medical products, the main user of which will be healthcare.
I think the work can be published in this Journal. However, some changes should be made before:
1- Introduction and background sections show some methods and examples, but does not mention theories and experiences of pioneering researches in emotional design and usability such as: Don Norman and Peter Desmet. I think that, if we are talking about this kind of aspects: usability, look & feel, and evaluation, the experiences of these authors can provide the basis for the study.
2-In this sens, I also think that these parameters could be described in a speciffic way, instead of generally. To differentiate the different parameters and set measurement shapes and aspects could be necessary.
3- Also in Figure 2, aspects such as who, ehich, were, when, etc appear, but they are not explained at the text. Specifically, "who" appears in the "look and Feel" and "Usability" levels. I do not know if it is an error, or this "aspect" is evaluated at both levels.
4- The method talks two times about focus group. However, their composition is not obsever: gender, age, occupation...
5- In addition, I think that the advantage or disadvantage of the use of this type of methodologies in the specific field of health should be considered in conclusion.
Author Response
Reviewer 2:
1- Introduction and background sections show some methods and examples, but does not mention theories and experiences of pioneering researches in emotional design and usability such as: Don Norman and Peter Desmet. I think that, if we are talking about this kind of aspects: usability, look & feel, and evaluation, the experiences of these authors can provide the basis for the study.
We accept the reviewer's recommendation. We elaborated on section 5.2 phase 2-usability, and section 5.3 phase 3- look and feel, adding refrences to Norman and Desmet work on this subject.
2-In this sens, I also think that these parameters could be described in a specific way, instead of generally. To differentiate the different parameters and set measurement shapes and aspects could be necessary.
We thank the reviewer for this comment. Section 6 provides a detailed case study presenting more specific aspects of the design process.
3- Also in Figure 2, aspects such as who, which, were, when, etc appear, but they are not explained at the text. Specifically, "who" appears in the "look and Feel" and "Usability" levels. I do not know if it is an error, or this "aspect" is evaluated at both levels.
We thank the reviewer for this comment. We elaborated as followed:
p4 l164: "The functionality phase focus on the technology part of the system, addressing a specific product's motive- ‘Why is this product needed?’, offering a technical solution- ‘What is the suggested solution?’, ‘How does it work?’. After establishing the foundation and setting an initial framework considering all technical and functional requirements, it is time to proceed to the usability phase, which is all about the users- ‘Who is the intended user?’, ‘Where and when will this product be in use?’, and what is the intended user-product interaction- meaning: ‘How is it going to be used?’. Having a primary concept for the intended device, the designer is able to proceed to the next phase: the look-and-feel of the product. This phase is again user-centered, hence deals with ‘Who is the user?’, ‘Which design language would fit?’. The final phase consists of an evaluation of the process as a whole."
4- The method talks two times about focus group. However, their composition is not obsever: gender, age, occupation...
We thank the reviewer for this comment. We elaborated as followed:
p8 l290: "In an on-going process, using various focus groups (medical professional teams, marketing directors, design students, etc.), adjectives were attached to each board… "
p9 l315: "FULE suggests using surveys and focus groups using intended users to complete the look-and-feel phase"
5- In addition, I think that the advantage or disadvantage of the use of this type of methodologies in the specific field of health should be considered in conclusion.
We thank the reviewer for this comment. We elaborated the conclusion section as followed:
p19 l634: “In this study, we sought to develop a novel design methodology referred to as FULE in an effort to facilitate an educated and efficient bias-free design process of complex medical devices. We presume that working according to the FULE guidelines could lead to a design problem's optimal solution. The new methodology addresses various aspects of the product: functionality, usability, and look-and-feel while considering safety and user experience. The case study demonstrates to that in the design of an autonomous medical device the FULE methodology provides the designer with tools to better select among design alternatives, and reduce bias and subjective decisions.”
Reviewer 3 Report
This article discusses the novel functionality, usability, appearance and evaluation (FULE) user-centered product design method proposed in this article. Its overall goal is to develop practical and beautiful products. Composed of several product design methods, this novel methodology of our design focuses on the roles and responsibilities of product designers. After defining the first three formative evaluation stages of product features, usability, and appearance, the aggregate evaluation stage not only evaluates the product, but also provides guidance on its implementation, marketing, and support. Based on a demonstration case study dedicated to the design of autonomous medical devices, it can be concluded that the FULE method can provide designers with tools to make better choices in design options and help reduce bias and subjective decision-making. , The research in this field and most of the content are narratives and instructions to the author as follows:
- The first three formative evaluation stages of product function, usability and appearance need to be clearly structured, the questions or research questions addressed in this article, and a clear agreement to show how the experimental part will answer the research questions; and adjust the content of the discussion based on the new results, what What is the meaning and conclusion of this article? Compared with previous studies on the same subject? , And the overall content is relatively incomprehensible to the practical theory of research.
- It is recommended that the research content explain clearly how to collect data, and practical decision-making tools should be added to analyze qualitative data. What is the overall relationship with the life cycle?
- Conducted six semi-structured interviews. What are the distribution of participants?
- Is the questionnaire survey conducted for reliability and validity analysis?
- How is this questionnaire asked and answered? The experienced medical staff of the medical center who used the system answered that two of them were eliminated because 455 surveys have not been fully completed. The rest includes 11 questionnaires filled out by 456 nursing staff and one questionnaire filled out by doctors. How can 455 experienced medical staff answer this question? Is the product consistent?
- As for the process method discussed in the research, the case study is an overview of its results. More case studies are needed for comparison. Three cases are used as comparative values. The process is applied with research theory, how to evaluate from the economic and environmental perspectives Case studies such as the model of a single packaging system are for the research needs more empirical verification and theoretically more complete research methods and theories.
- Are all the documents cited in the manuscript relevant to this paper, if there is no need to make adjustments?
Author Response
Reviewer 3:
- The first three formative evaluation stages of product function, usability and appearance need to be clearly structured, the questions or research questions addressed in this article, and a clear agreement to show how the experimental part will answer the research questions; and adjust the content of the discussion based on the new results, what What is the meaning and conclusion of this article? Compared with previous studies on the same subject? , And the overall content is relatively incomprehensible to the practical theory of research.
We thank the reviewer for this comment. We elaborated the conclusion section as followed:
p19 l634: “In this study, we sought to develop a novel design methodology referred to as FULE in an effort to facilitate an educated and efficient bias-free design process of complex medical devices. We presume that working according to the FULE guidelines could lead to a design problem's optimal solution. The new methodology addresses various aspects of the product: functionality, usability, and look-and-feel while considering safety and user experience. The case study demonstrates to that in the design of an autonomous medical device the FULE methodology provides the designer with tools to better select among design alternatives, and reduce bias and subjective decisions.”
- It is recommended that the research content explain clearly how to collect data, and practical decision-making tools should be added to analyze qualitative data. What is the overall relationship with the life cycle?
- Conducted six semi-structured interviews. What are the distribution of participants?
We thank the reviewer for these two comments. The information is provided in section 5.2 usability:
p6 l231: "The tools we use for the learning stage are varied and may differ from project to project. Still, they will always include examining an existing/ similar system (if any), interviews with the various users, and observations in the work environment and similar procedures. Additional tools may be anonymous questionnaires, pollination (searching for ideas in tangential or other content worlds), a literature review, and consulting with experts from various fields. After collecting the data, we perform the analysis phase by building task sequence models and usage scenarios. These models are used for risk analysis, required information and interface analysis (depending on system requirements and capabilities, and user needs), and conflict analysis. The methodology uses graphical and visual tools to represent different product features and requirements.
Sometimes at this stage, it is found that some requirements conflict. FULE methodology encourages decision making using a visual representation of the alternatives using the Harris profile [21] (our case study demonstrates the use of this tool in Section 6.2) or by creating different graphs presenting the contradiction."
And in section 6.6 Data Gathering:
p15 l488: "Six semi-structured interviews were conducted. Participants included six volunteering nurses, three males and three females who had experienced the AG100s system. The interviews took place at Rambam Medical Center. The participants were asked to share their knowledge, opinions, and perceptions regarding ventilator-associated pneumonia (VAP), the AG100s system's usability and appearance, and autonomic medical systems in general. All interviews were recorded and transcribed in full. The data were analyzed thematically.
- Is the questionnaire survey conducted for reliability and validity analysis?
Due to the small number of participants in each survey we were not able to conduct reliability and validity analysis. As explained in section 6.6 Data Gathering:
p19 l602: "This is a common situation due to the fact that we were evaluating a product for the first time with real users, employing a small group of participants."
- How is this questionnaire asked and answered? The experienced medical staff of the medical center who used the system answered that two of them were eliminated because 455 surveys have not been fully completed. The rest includes 11 questionnaires filled out by 456 nursing staff and one questionnaire filled out by doctors. How can 455 experienced medical staff answer this question? Is the product consistent?
We thank the reviewer's comment. We elaborated as followed:
p15 l484 "The evaluation phase was conducted at the ICU cardiac surgery units at Rambam Medical Center among trained nursing staff who were using the new AnapnoGuard system for six months."
- As for the process method discussed in the research, the case study is an overview of its results. More case studies are needed for comparison. Three cases are used as comparative values. The process is applied with research theory, how to evaluate from the economic and environmental perspectives Case studies such as the model of a single packaging system are for the research needs more empirical verification and theoretically more complete research methods and theories.
We thank the reviewer for the encouraging feedback.
- Are all the documents cited in the manuscript relevant to this paper, if there is no need to make adjustments?
We thank the reviewer for his comment. However, we think that the references that we present are important and would benefit the readers of our paper.
Round 2
Reviewer 1 Report
The revised version of the article has been substantially improved by providing a clearer scope of the contribution and more relevant information regarding the implementation of the proposed methodology.
The authors failed however to clarify the assessment of the methodology. What they discuss in Sections 6 and 7 is the product development and evaluation according to FULE, but not the assessment of FULE itself. This still remains problematic with respect to the claims of objectivity and cognitive bias reduction of FULE.
At p.4, l.138-142, the authors seem to state that they first designed the methodology, by setting some guidelines and principles first, and then applied it in the case study.
Which are these guidelines and principles?
How were the tools and methods selected (l.145)?
Was there some experimental assessment or comparison of the candidate tools and methods?
My still concern is mainly logical, that is I don't see evidence that the evaluation of a project equates to the evaluation of the process, as it is stated at p.20, l.638-640.
Therefore, I recommend to soften the claims regarding the efficiency of FULE in providing the "designer with tools to better select among design alternatives and reduce bias and subjective decisions".
In order to do that FULE should be tested in numerous other projects, and by design teams other than the authors, and according some shared protocols.
--------------
Minor comments
--------------
- p.3, l.104: "several design projects", only the medical device case study is analysed.
- p13, l.444: the three concepts are not discussed, a discussion could have been useful to provide evidence of how the chosen method in this FULE stage can help solving problems and potential conflicts. This implies though a discussion of the emerging problems and conflicts and a demonstration of method effectiveness, by comparing it with other approaches.
Author Response
Reviewer 1:
- The authors failed however to clarify the assessment of the methodology. What they discuss in Sections 6 and 7 is the product development and evaluation according to FULE, but not the assessment of FULE itself. This still remains problematic with respect to the claims of objectivity and cognitive bias reduction of FULE.
Section 6 – Case study (line 341) is an example to the implementation of the FULE methodology. This successful implementation presents the improved outcomes results from using this methodology for product development. This is the best assessment we could have to the methodology.
- At p.4, l.138-142, the authors seem to state that they first designed the methodology, by setting some guidelines and principles first, and then applied it in the case study.
Which are these guidelines and principles?
How were the tools and methods selected (l.145)?
Was there some experimental assessment or comparison of the candidate tools and methods?
The guidelines of each phase are detailed in its section.
Tools and methods were selected and developed as a result of analysis conducted at the end of every design project, as mentioned in line 59:
“Using failure analysis at the end of every design project helped us analyze obstacles, find solutions, and determine rules. These were then applied in the next projects. The result was a clear guide that leads the designer through the whole procedure, starting at the beginning of a process and ending only after the product is already in use by real users.”
- My still concern is mainly logical, that is I don't see evidence that the evaluation of a project equates to the evaluation of the process, as it is stated at p.20, l.638-640.
Therefore, I recommend to soften the claims regarding the efficiency of FULE in providing the "designer with tools to better select among design alternatives and reduce bias and subjective decisions".
In order to do that FULE should be tested in numerous other projects, and by design teams other than the authors, and according some shared protocols.
We thank the reviewer for this comment and made changes to the sentence in line 637-639 to reflect the fact that we can’t quantitively compare tools. The update sentence is:
We assume that working according to the FULE guidelines could lead to the design of better solutions for design challenges.
However, we think that the case study demonstrates how the tools we present in the paper result in better design, and this claim is demonstrated in few palaces along the paper. For example:
Line 344: “The design process was initiated even though the company already had a first functional prototype aiming to evaluate the reduction of VAP occurrences. Usability issues that were not considered during the first prototype development process resulted in operational challenges in hospitals, and nursing staff refused to handle the system due to its complexity. Hence, this project's goal was to improve its usability, along with achieving acceptance and trust. We used FULE in the development of the second generation of the product in an effort to improve user interaction.”
Line 139: “We then demonstrate the FULE's advantages through a medical device case study where FULE was used to develop the second generation of a medical product that was complex and had usability difficulties. The case study helped us evaluate the FULE methodology through feedback that was collected from medical staff in two hospitals.”
- 3, l.104: "several design projects", only the medical device case study is analysed.
In this paper we present the methodology, through examples from several design projects, and use the case study as an example to the implementation of this methodology. Therefore, only the case study is analysed in detail, as stated in section 3. Aim:
Line 134: “In this paper, we demonstrate some of the FULE phases and tools in several design projects. In addition, we present a case study of the full design process of a complementary autonomic breathing system by Hospitech®. “
- p13, l.444: the three concepts are not discussed, a discussion could have been useful to provide evidence of how the chosen method in this FULE stage can help solving problems and potential conflicts. This implies though a discussion of the emerging problems and conflicts and a demonstration of method effectiveness, by comparing it with other approaches.
We thank the reviewer for this comment. We elaborated as followed:
Line 437: “Observations and interviews' outcomes were used to help us design three different concepts of the new device. Top three suggestions for the system location were: 1. Attached to patient bed, 2. Connected to the back wall, and 3. Using designated pole. We used Harris's profile [22] to help us visualize each design concepts' strengths and weaknesses according to our declared requirements (Figure 10).
Figure 10: Using the Harris profile for concept comparison in our case study.
Using this four-scale scoring with a graphic representation helped us easily compare between our three options (derived from conducted interviews and observations) and chose to design a designated pole that fit the design requirements in the best way.
Reviewer 3 Report
This article discusses the novel functionality, usability, appearance and evaluation (FULE) user-centered product design method proposed in this article. Its overall goal is to develop practical and beautiful products. Composed of several product design methods, this novel methodology of our design focuses on the roles and responsibilities of product designers. After defining the first three formative evaluation stages of product features, usability, and appearance, the aggregate evaluation stage not only evaluates the product, but also provides guidance on its implementation, marketing, and support. Based on a demonstration case study dedicated to the design of autonomous medical equipment, it can be concluded that the FULE method can provide designers with tools to make better choices in design options and help reduce bias and subjective decision-making. Most of the research in this field is narrative. The author’s content still needs to be strengthened to explain whether the medical field is really valid.
Passed after revision in this article.
Author Response
Reviewer 3:
This article discusses the novel functionality, usability, appearance and evaluation (FULE) user-centered product design method proposed in this article. Its overall goal is to develop practical and beautiful products. Composed of several product design methods, this novel methodology of our design focuses on the roles and responsibilities of product designers. After defining the first three formative evaluation stages of product features, usability, and appearance, the aggregate evaluation stage not only evaluates the product, but also provides guidance on its implementation, marketing, and support. Based on a demonstration case study dedicated to the design of autonomous medical equipment, it can be concluded that the FULE method can provide designers with tools to make better choices in design options and help reduce bias and subjective decision-making. Most of the research in this field is narrative. The author’s content still needs to be strengthened to explain whether the medical field is really valid.
Passed after revision in this article.
The methodology presented in this paper is focused on medical devices. This is a growing field that although focused on functionality has started to pay attention to design in recent years. This is also why the case study that was selected as an example to the implementation of the FULE methodology is a medical device.
This manuscript is a resubmission of an earlier submission. The following is a list of the peer review reports and author responses from that submission.
Round 1
Reviewer 1 Report
The paper describes a methodology for product design called FULE - Functionality, Usability, Look-and-Feel, and Evaluation.
The authors describe the methodology in a case study of the design of a medical ventilator.
I think that the description of the design process of medical devices is relevant and timely and I think that an in-depth analysis of the design process for the VAP would make an interesting paper. However, the present paper is intended as a contribution of a new design methodology. As such, I think it has a few major deficiencies.
The first is that the need for a new methodology is not clearly motivated. The Introducation does not clearly explain why the new FULE methodology is needed. Given the various possible methodologies for interaction design, why is FULE relevant?
The second is that there is not enough analysis of existing methodologies and how FULE compares to them. What are the weak points in existing methodologies? What does FULE add that existing methodologies do not provide?
The third is that the design of the proposed methodology is not explained at all. The method section states only that "we first set out to define guidelines and principles to the FULE user-centered methodology, thereby engaging in the process of theory building." It is not clear what this means. How did the new methodology come to be? It seems that FULE was just the result of a single case where it was applied but it is not even clear if the authors derived the methodology after the fact (after the case study) or intentionally applied in the case study. In any case, a single application of a new methodology is hardly evidence of its relevance.
The fourth is that the proposed methodology is not evaluated. It is described as applied in a case study but this does not allow us to determine if a different methodology would have resulted in a worse design, nor does it allow us to determine if applying the methodology in a different case will also be a success.
In addition, the paper has a few other issues.
In the Background, it is not clear the relevance of section 1.2 Autonomous systems (which should be numbered as 2.2), 1.3, or 1.4. Trust is mentioned a few times, but it is not clear how the proposed methodology actively addresses it.
FULE is depicted in a pyramid and authors state that "there is a clear hierarchy". I could not understand the meaning of this hierarchy.
Phase 2: Usability is not sufficiently well described. Four phases are mentioned, but not described.
I could not understand the relationship between the Look&Feel phase and desirability, nor why authors proposed the separation of Usability and Look&Feel. There seems to be a naive association between user experience and simple aesthetical properties like coloour and fonts, which is not explained.
It is hard to accept the proposed methodology as a good methodology to follow generally. It seems a sequential methodology with a single evaluation phase at the end. But what happens if the evaluation reveals flaws only at the end of the cycle? This means that a lot of work and effort have already been expended.
I find it surprising that no mention of interaction design process is made in the paper.
Reviewer 2 Report
The topic is interesting that the approach may provide a systematic guideline for product design. However, the case of autonomous medical device is professional that its function is the most important criteria. There is major hesitation on the aspects of "look and feel" and "usability" being included in the model.
The quality and size of figures (e.g. Figure 4, 5, 7 and 8) may be difficult to understand. The current images of medical devices without details or description may also be difficult to link with the assessment of "pleasing to the eye", "friendly", "medical" and "reliable", etc.